# Digital Competence of University Teachers of Social and Legal Sciences from a Gender Perspective

Sergio Cored Bandrés *, Marta Liesa Orús , Sandra Vázquez Toledo , Cecilia Latorre Cosculluela and Silvia Anzano Oto

Department of Educational Sciences, University of Zaragoza, 50009 Zaragoza, Spain; martali@unizar.es (M.L.O.); svaztol@unizar.es (S.V.T.); clatorre@unizar.es (C.L.C.); sanzano@unizar.es (S.A.O.)
* Correspondence: scoban@unizar.es

**Abstract:** Higher Education institutions must respond to the major challenges posed by the technological transformations of recent years. For this, one of the key aspects is that the teachers working in them are trained to incorporate these technologies in teaching–learning processes, which requires them to be digitally competent. To analyse how gender relates to the level of competence of teachers, the types of ICT tools that they use, and their attitudes towards their implementation in teaching–learning processes, this quantitative research was carried out with the participation of 121 university teachers from the Autonomous Community of Aragon, all of them from the area of Social and Legal Sciences. The results show a differentiated profile between men and women in terms of their attitudes, beliefs, and self-perception regarding their management skills and effectiveness.

**Keywords:** digital competence; social and legal sciences; higher education; university professors; gender



## 1. Introduction

In recent years, society has undergone a profound technological transformation that has permeated all areas of society, including education [1–3]. The incorporation of Information and Communication Technologies (from now on, ICTs) has diametrically changed the teaching–learning and communication processes, generating a new reality. Higher Education institutions cannot remain oblivious to all these transformations and must provide a coherent and effective response to current demands, which can sometimes be disorientating. One of the aspects in which universities must undoubtedly make progress is the development of the digital competence of their teaching staff and their use of didactic applications, through the introduction of technologies in their classrooms as forms of communication and access to information [4–6].

In the same way, it is necessary to promote the training of their teaching staff in the utilisation of media through the use of online applications for didactic purposes that allow them to interact with the media critically and creatively, produce new content, or take advantage of existing content [6–12]. Today's university students are digital natives [13], and their ways of interacting and learning are markedly different from that of past generations [14,15]. Higher Education environments are demanding a reformulation of university teaching that takes into account the symbiotic relationship established between students and ICT [16,17] since their learning process is strongly influenced by technologies [18,19]. This conception is not new and has already been supported by authors such as Castell [20], who stated in 2000 that society could not be understood or represented without bearing in mind its technological tools.

Teachers in the current university system must not only be technologically literate but also digitally competent [21]. Information technologies allow them "to be able to use more active and stimulating methodologies, and to get more involvement from students in their teaching–learning process" [22] (p. 132). In the same sense, the contributions of Cabero

and Marín [23] point out that the use of ICT in teaching–learning processes is not a sine qua non for quality and innovative teaching, but it favours the employment of more active, attractive, and motivating methodologies [22].

As a result of all of the above, the concept of digital competence emerges, which is defined as the critical and safe use of Information Society Technologies for work, leisure, and communication, and is one of the eight key competences necessary for Lifelong Learning [24]. Moreover, let us not forget that it is listed as a key competence in the 2030 Sustainable Development Goals.

In universities, digital competence is present in the verification reports of different university degrees. However, few degree programmes include one or more specific subjects related to it, as this digital competence is dealt with transversally in most degrees. Fernández et al. [25] carried out an investigation in the Bachelor's Degrees of the macro-area of Social and Legal Sciences at the University of Malaga. Among their results, it was found that students had had to acquire this type of competence autonomously, given that they had not received systematic training for it. This reality seems to go against what educational experience indicates, which corroborates the importance given to acquiring this competence both for the good performance of educational tasks and academic progress and for the subsequent incorporation into the labour market [25]. In any case, the role played by ICT in our society should not be questioned, nor should the need to address ICT in education.

### 1.1. Competence and Attitude of University Teachers

The inclusion of ICT in university classrooms is conditioned by the digital competence of the professors that implement them [26–28]. A large number of studies have highlighted the importance of addressing digital competence in universities [26–30] and the need to improve the digital competence of university teaching staff [23,31]. This need for change is reflected in the results of some studies which, based on the three fundamental components of digital competence (attitude, knowledge, and ability to use technologies), confirm that technology is generally used simply as a means of obtaining information (through search engines such as Google) and processing information (Word, PowerPoint, etc.) [27]. Other studies show that teachers lack the necessary training, time, and resources to integrate ICT into everyday university teaching [4,32,33].

Teachers' perceptions of ICT play an important role [34], as teachers are the ones who put ICT into practice. They decide which resources are used in the classroom and how often, so they incorporate their views into the curriculum design [35].

However, there is a large body of previous literature that shows a high correlation between knowledge of ICT and the use of ICT by university teachers [5,31]. In other words, more training in ICT increases teachers' digital competence, which is fundamentally related to self-efficacy and the perception of the impact of ICT in education [36]. Therefore, the development of digital competence could be the goal to be achieved in order to bring university teachers closer to a coherent integration of ICT in their teaching [37].

We cannot ignore the fact that the inclusion of ICT in universities has posed great challenges to the academic activities of teachers. The great potential of these digital tools has affected the education that students receive, as it has led to the emergence of new teaching and learning strategies in different disciplines [4,5,27]. This is because technology has transformed the current learning environment from a traditional teacher-centred one to a student-centred one, where the teacher has become a guide and the student has moved from being a passive recipient of information to an active participant in his or her own learning [38,39].

Although the incorporation of ICT into teaching and learning processes at the university level is a great challenge, it brings clear advantages for teachers, students, and the universities themselves [26,40]. Among other aspects, ICTs support students and teachers and improve communication in educational contexts [41]. However, even though there are many reasons to use ICT at universities [22], this is not the only element that ensures

quality and innovative teaching. Nonetheless, it does allow teachers to incorporate their point of view in the educational curriculum when making decisions about the inclusion of these tools in the classroom [35]. The integration of ICT into the teaching–learning process requires a high amount of training, updating and dedication on the part of university professors [23]. We must add to all this the perceptions and attitudes that university teaching staff have towards this matter. If they have a positive attitude, it will be easier to incorporate ICT as an essential tool in the teaching–learning process [34,42]. For this reason, studies such as the present one are of great relevance, as they serve as a basis for evaluation and reflection in order to subsequently define lines of research and action that favour the improvement of attitudes and perceptions affecting the teaching–learning process.

### 1.2. Digital Competence of University Professors According to Gender

Gender differences in the performance of technological competence is a widely studied topic and the results provided by the literature have often been contradictory. Some research [43] shows that men and women are equally competent, but other studies suggest that men are more skilled [44–46]. In the latter cases, there is a digital gap affecting women in terms of ICT-use and competence. Some studies have analysed technological competence according to the gender variable in which women are shown to be more competent [47]. This is the case of the research that Araiza and Pedraza [48] carried out with university teachers. The opinions of the participants on the uses of ICT in the classroom and their digital competencies were assessed, and significant differences were found between men and women. Women teachers valued positively with teaching competence in ICT and the factors that determine its incorporation in the teaching process at the Higher Education level. Regarding the integration of ICT in disciplinary subjects as an innovation for educational improvement, women teachers also gave a higher value to the usefulness of being able to evaluate students' learning performance, and as a means to improve the development of their classes.

These discrepancies and differences in the results of previous research highlight the importance of the present study, which intends to delve deeper into this issue, focusing on the gender variable as a relevant aspect to investigate.

Based on all that has been said so far, and in order to dig deeper into this topic, the following hypotheses are proposed:

- Women perceive higher self-efficacy in the use of technologies than men;
- Both genders have a positive attitude towards the use of ICT;
- Women rate ICT more positively as a tool for educational support than men;
- Both genders value the effectiveness of ICT for the development of 21st century skills.

In relation to these hypotheses, the present study has three main objectives. According to the first one, the aim is to study the self-efficacy perceived by university teachers according to gender in relation to the use of technology as a tool to improve the teaching–learning process, as well as to observe possible significant differences between the management and effectiveness of various ICT tools.

The second objective is to compare the attitudes of men and women teachers regarding their predisposition towards the use of ICT and their behaviour in the use of technologies to support the educational process. The last and third objective is to analyse the gender differences in university teachers' beliefs about the effectiveness of ICT in the development of 21st century competencies.

The specific objectives for each of the general objectives of this study are:

- To find significant differences according to teachers' gender in relation to their perceived self-efficacy in the use of technologies;
- To analyse differences according to teachers' gender in their assessment of technologies as educational aids;
- To analyse the effectiveness of ICT in developing 21st century skills and to see if there are gender differences in this respect.

## 2. Materials and Methods

This study follows a cross-sectional quantitative approach under a descriptive and non-experimental survey design based on a questionnaire, the most common technique for data collection in the field of Social Sciences [49]. This methodological approach allows us to explore and describe the target of the study.

### 2.1. Participants and Sample

The study population consisted of 121 university teachers from the Autonomous Community of Aragon belonging to the branch of knowledge of Social and Legal Sciences (see Table 1). The participating faculties were the three Faculties of Education of the three provinces of Aragon, the Faculty of Social Work, the Faculty of Business, and the Faculty of Health and Sport among others. All of them belong to the University of Zaragoza.

**Table 1.** Characterisation of the sample (N = 121).

| VARIABLE | N | % |
|---|---|---|
| **Gender** | | |
| Men | 60 | 49.6 |
| Women | 61 | 50.4 |
| **Age (M = 46.44; DT=11.051)** | | |
| 24–40 years | 35 | 28.9 |
| 41–55 years | 68 | 56.2 |
| 56–73 years | 18 | 14.9 |
| **Professional Profile** | | |
| Civil servant teaching staff (professors and tenured professors) | 52 | 43 |
| Non-civil servant teaching staff (Contracted Ph.D., Assistant Ph.D., and Interim) | 33 | 27.2 |
| Associate Professors | 36 | 29.8 |
| **Years of teaching experience (M=15.65; DT=11.682)** | | |
| 1–15 years | 68 | 56.2 |
| 16–30 years | 37 | 30.6 |
| 31–45 years | 16 | 13.2 |

The sampling was accidental and non-probabilistic [50]. The questionnaire was emailed by the researchers to the deans of each of the faculties that are part of the University of Zaragoza for dissemination, and was answered freely and voluntarily.

Thus, the sample was made up of university professors who agreed to participate in the research. A total of 60 participants were men (49.6%) and 61 were women (50.4%). In terms of age, they could be divided into three different ranges: 28.9% of them were between 24 and 40 years old, 56.2% between 41 and 55, and 14.9% between 56 and 73, the average age of all of them being 46.44 years. A total of 43% of the participants were civil servant professors (professors and tenured professors), 27.2% were non-tenured professors (Contracted PhD, Assistant PhD, and Interim), and 29.8% were associate professors. Finally, 56.2% had 1 to 15 years of teaching experience in Higher Education, while 30.6% had 16 to 30 years, and 13.2% had between 31 and 45 years of experience.

### 2.2. Data Collection Instrument

The questionnaire used to obtain the data was structured into four distinct blocks. Blocks A and D were taken from the SABER-TIC questionnaire by Taquez et al. [51]. Block A, entitled "ICT management", presented 18 items in two Likert-type scales from 1 to 10. ICT management referred to the decision-making and procedures to be used in order to make the best possible use of these tools. Teachers assessed the perceived management and effectiveness of different ICT tools. Effectiveness referred to the ability of something to produce the desired effect. Block D ("Competences for the 21st century") was composed of 18 items on a Likert-type scale from 1 to 10, with 1 being "totally disagree" and 10 "totally agree". Finally, Block B ("Attitudes of predisposition towards the use of ICT"), consisting

of 8 items, and Block C ("Behaviour in the use of ICT to support the educational process"), consisting of 11 items, were obtained from the Agreda et al. [52] questionnaire.

The reliability of this instrument was assessed through Cronbach's Alpha and a result of 0.971 was obtained, which is understood as an acceptable internal consistency value [53].

*2.3. Procedure*

The participating university teachers were contacted by means of an email sent from the administrative offices of the participating university, using distribution lists. The content of the email informed the teachers about the objectives of the study and asked them to participate voluntarily and anonymously in it. In addition, the web link leading to the online platform supporting the questionnaire was attached. After collection, the data were cleaned and entered into a database.

The data obtained were coded and analysed with the SPSS statistical software (version 26). First, the normality of the data was checked by performing the one-sample K-S (Kolmogorov–Smirnov) test. A $p > 0.05$ was obtained, indicating that the distribution of the data was normal. After checking this, an analysis of the means and standard deviations of the items that made up the questionnaire was carried out. At the same time, a comparison was made of these indicators for each dimension according to the gender of the participants using Student's *t*-test for independent samples. In addition, in order to detect significant differences between ICT management and effectiveness, Student's *t*-test for related samples was applied.

## 3. Results

The results derived from the present study were structured around the six dimensions of analysis: ICT management, degree of effectiveness of ICT in teaching–learning processes, comparisons between the degree of ICT management and effectiveness, attitudes of predisposition to ICT-use, teachers' use of ICT as support in the educational process, and the effectiveness of ICT for the development of 21st century competencies.

Firstly, as can be seen in Table 2, the results on the use of ICT tools were variable in both genders. In relation to men, the types of tools that registered a higher degree of management were word processors and spreadsheets (M = 8.15; SD = 1.783), the basic components that referred to peripheral elements or storage (M = 8.13; SD = 1.808), and the web and its basic tools such as mail or distribution lists (M = 7.95; SD = 2.103). Conversely, there were seven kinds of ICT tools for which the score obtained was less than five. These included social bookmarking tools for sharing information (M = 1.58; SD = 2.424), tools for creating QR codes (M = 3.33; SD = 3.467), Personal Learning Environments (M = 3.70; SD = 3.201), and online publishing tools such as Pinterest (M = 3.88; SD = 3.289).

Concerning women, the tools in which a higher degree of proficiency was pointed out were management platforms (M = 8.46; SD = 1.467), basic web tools (M = 8.33; SD = 1.938), and basic ICT components (M = 8.10; SD = 1.930). Additionally, only the management of three types of tools received a rating below five for women. Specifically, those related to social bookmarking (M = 1.61; SD = 2.551), tools for creating QR codes (M = 3.56; SD = 3.139), and Personal Learning Environments (M = 4.84; SD = 2.876).

Comparing both genders, men scored higher in only two tools (basic components and operating systems), although these differences were not significant. In the remaining cases, the participating women scored higher and, in four of them, the differences were significant (*p*-value < 0.05). Specifically, in the management of social networks ($p = 0.006$), in the effective search for information on the web ($p = 0.016$), in management platforms ($p = 0.042$), and in Personal Learning Environments ($p = 0.042$).

**Table 2.** Comparison of the use of ICT tools by gender.

| ÍTEMS | Gen. | Driving (Ma) | | Test T | |
| --- | --- | --- | --- | --- | --- |
| | | M | DT | p | Contrast |
| 1. Basic components of ICTs (storage, peripheral elements, etc.). | Men | 8.13 | 1.808 | 0.918 | Mm1 = Mw1 |
| | Women | 8.10 | 1.930 | | |
| 2. Operating systems and their management (word processors, spreadsheets, etc.). | Men | 8.15 | 1.783 | 0.300 | Mm2 = Mw2 |
| | Women | 7.77 | 2.201 | | |
| 3. Web and basic tools (mail, browsers and distribution lists). | Men | 7.95 | 2.103 | 0.306 | Mm3 = Mw3 |
| | Women | 8.33 | 1.938 | | |
| 4. Social networks | Men | 5.28 | 3.400 | **0.006 *** | Mm4 < Mw4 |
| | Women | 6.80 | 2.581 | | |
| 5. Resources through Web 2.0 applications (blogs, Wikis, Forums, etc.). | Men | 5.23 | 2.919 | 0.491 | Mm5 = Mw5 |
| | Women | 5.59 | 2.765 | | |
| 6. Storage within cloud environments (Google Drive, Dropbox, etc.) | Men | 6.93 | 3.058 | 0.140 | Mm6 = Mw6 |
| | Women | 7.67 | 2.379 | | |
| 7. Social bookmarking and content syndication to share information (Delicious, Mister Wong, FeedReader, etc.). | Men | 1.58 | 2.424 | 0.959 | Mm7 = Mw7 |
| | Women | 1.61 | 2.551 | | |
| 8. Management platforms (Moodle, Blackboard, etc.) | Men | 7.88 | 1.606 | **0.042 *** | Mm8 < Mw8 |
| | Women | 8.46 | 1.467 | | |
| 9. Device protection software | Men | 4.95 | 3.377 | 0.249 | Mm9 = Mw9 |
| | Women | 5.61 | 2.836 | | |
| 10. Databases and information search thesauri | Men | 7.30 | 1.977 | 0.169 | Mm10 = Mw10 |
| | Women | 7.79 | 1.890 | | |
| 11. QR code creation tools | Men | 3.33 | 3.467 | 0.710 | Mm11 = Mw11 |
| | Women | 3.56 | 3.139 | | |
| 12. Personal Learning Environments | Men | 3.70 | 3.201 | **0.042 *** | Mm12 < Mw12 |
| | Women | 4.84 | 2.876 | | |
| 13. Collaborative use of ICT | Men | 5.57 | 3.011 | 0.169 | Mm13 = Mw13 |
| | Women | 6.31 | 2.913 | | |
| 14. Development of materials through presentations and multimedia | Men | 6.42 | 2.770 | 0.342 | Mm14 = Mw14 |
| | Women | 6.90 | 2.827 | | |
| 15. Knowledge of copyrights and intellectual property | Men | 4.53 | 3.218 | 0.150 | Mm15 = Mw15 |
| | Women | 5.33 | 3.275 | | |
| 16. Bibliographic managers (Zotero, Refworks, etc.) | Men | 4.53 | 3.218 | 0.181 | Mm16 = Mw16 |
| | Women | 5.33 | 3.275 | | |
| 17. Efficient search and discrimination of relevant information on the web. | Men | 6.65 | 3.080 | **0.016 *** | Mm17 < Mw17 |
| | Women | 7.84 | 2.177 | | |
| 18. Online publishing tools (Pinterest, Instagram, SlideShare, etc.). | Men | 3.88 | 3.289 | 0.050 | Mm18 = Mw18 |
| | Women | 5.08 | 3.373 | | |

Gen. = gender; Mm = mean men; Mw = mean women; * $p < 0.05$: significant change.

As for the degree of effectiveness of different ICT tools in improving teaching–learning processes (Table 3), the results were also disparate. The participating men considered management platforms (M = 8.03; SD = 2.277), basic ICT components (M = 7.85; SD = 1.745), and word processors, spreadsheets, etc. (M = 7.92; SD = 1.788) to be the most effective tools. In contrast, six types of technology scored below five. These lowest scores were for social bookmarking (M = 2.30; SD = 2.493), QR code creation tools (M = 2.82; SD = 2.765), and device protection software (M = 3.68; SD = 3.234). For the women, the most highly rated tools were the effective search and discrimination of relevant information on the web

(M = 8.43; SD = 2.004), management platforms (M = 8.39; SD = 2.304), and web and basic tools such as mail (M = 8.20; SD = 2.023). The ICT tools with a score of less than five were only social bookmarking (M = 3.07, SD = 2.869) and tools for creating QR codes (M = 3.84, SD = 2.740).

If we contrast the results for the two genders, the score was higher for women than for men in all the items. Six of these scores were significantly higher ($p < 0.05$): device protection software ($p = 0.002$), social networking ($p = 0.006$), resources through web 2.0 applications such as blogs or forums ($p = 0.019$), cloud storage ($p = 0.021$), effective web search ($p = 0.037$), and tools for creating QR codes ($p = 0.044$).

**Table 3.** Comparison of the effectiveness of different ICT tools by gender.

| ÍTEMS | | Effectiveness (Ef) | | Test T | |
|---|---|---|---|---|---|
| | Gen. | M | DT | *p* | Contrast |
| 1. Basic components of ICTs (storage, peripheral elements . . . ). | Men | 7.85 | 1.745 | 0.659 | Mm1 = Mw1 |
| | W. | 8.00 | 1.975 | | |
| 2. Operating systems and their management (word processors, spreadsheets, etc.). | Men | 7.92 | 1.788 | 0.508 | Mm2 = Mw2 |
| | W. | 8.13 | 1.765 | | |
| 3. Web and basic tools (mail, browsers and distribution lists) | Men | 7.65 | 2.138 | 0.151 | Mm3 = Mw3 |
| | W. | 8.20 | 2.023 | | |
| 4. Social networks | Men | 4.27 | 2.857 | **0.006 *** | Mm4 < Mw4 |
| | W. | 5.69 | 2.760 | | |
| 5. Resources through web 2.0 applications (blogs, Wikis, Forums, etc.) | Men | 5.25 | 3.062 | **0.019 *** | Mm5 < Mw5 |
| | W. | 6.46 | 2.494 | | |
| 6. Storage within cloud environments (Google Drive, Dropbox, etc.) | Men | 6.23 | 3.050 | **0.021 *** | Mm6 < Mw6 |
| | W. | 7.43 | 2.520 | | |
| 7. Social bookmarks and content syndication to share information (Delicious, Mister Wong, FeedReader, etc.) | Men | 2.30 | 2.493 | 0.120 | Mm7 = Mw7 |
| | W. | 3.07 | 2.869 | | |
| 8. Management platforms (Moodle, Blackboard, etc.) | Men | 8.03 | 2.277 | 0.389 | Mm8 = Mw8 |
| | W. | 8.39 | 2.304 | | |
| 9. Device protection software | Men | 3.68 | 3.234 | **0.002 *** | Mm9 < Mw9 |
| | W. | 5.43 | 2.747 | | |
| 10. Information search databases and thesauri | Men | 7.77 | 2.037 | 0.641 | Mm10 = Mw10 |
| | W. | 7.95 | 2.291 | | |
| 11. Tools for creating QR codes | Men | 2.82 | 2.765 | **0.044 *** | Mm11 < Mw11 |
| | W | 3.84 | 2.740 | | |
| 12. Personal Learning Environments | Men | 4.87 | 3.372 | 0.060 | Mm12 = Mw12 |
| | W | 5.95 | 2.901 | | |
| 13. Use of ICT in a collaborative way | Men | 6.35 | 3.102 | 0.136 | Mm13 = Mw13 |
| | W. | 7.13 | 2.611 | | |
| 14. Preparation of materials through presentations and multimedia | Men | 7.12 | 2.558 | 0.312 | Mm14 = Mw14 |
| | W. | 7.59 | 2.571 | | |
| 15. Knowledge of copyright and intellectual property | Men | 6.13 | 2.890 | 0.085 | Mm15 = Mw15 |
| | W. | 7.02 | 2.705 | | |
| 16. Bibliographic managers (Zotero, Refworks, etc.) | Men | 5.67 | 3.198 | 0.598 | Mm16 = Mw16 |
| | W. | 5.95 | 2.177 | | |
| 17. Effective search and discrimination of relevant information on the web | Men | 7.48 | 2.855 | **0.037 *** | Mm17 < Mw17 |
| | W. | 8.43 | 2.004 | | |
| 18. Online publishing tools (Pinterest, Instagram, SlideShare, etc.) | Men | 4.12 | 3.157 | 0.051 | Mm18 = Mw18 |
| | W. | 5.25 | 3.150 | | |

Gen. = gender; Mm = mean men; Mw = mean women; * $p < 0.05$: significant change.

Regarding the comparison between ICT management and effectiveness (Table 4), the Student *t*-test for related samples revealed significant differences in some ICT elements for both men and women. In the case of men, this difference occurred in 10 of the tools. In three of them, management scores were higher than effectiveness scores: social networks ($p = 0.002$), cloud storage ($p = 0.041$), and device protection software ($p = 0.006$). On the contrary, the opposite was true for the rest of the tools included in the dimension. With respect to women, these significant differences were perceived in eight of the items (seven of them coinciding with the results for men). Only social networks ($p = 0.004$) obtained higher scores in their management than in their effectiveness. In the rest of the cases, effectiveness received a higher score.

**Table 4.** Comparison between ICT management and efficacy by gender.

| ÍTEMS | Gen. | Driving (Ma) | | Effectiveness (Ef) | | Test T | |
|---|---|---|---|---|---|---|---|
| | | M | DT | M | DT | p | Contrast |
| 1. Basic components of ICTs (storage, peripheral elements, etc.) | Men | 8.13 | 1.808 | 7.85 | 1.745 | 0.273 | Ma1 = Ef1 |
| | W. | 8.10 | 1.930 | 8.00 | 1.975 | 0.685 | Ma1 =Ef1 |
| 2. Operating systems and their management (word processors, spreadsheets, etc.). | Men | 8.15 | 1.783 | 7.92 | 1.788 | 0.399 | Ma2 = Ef2 |
| | W. | 7.77 | 2.201 | 8.13 | 1.765 | 0.107 | Ma2 = Ef2 |
| 3. Web and basic tools (mail, browsers and distribution lists) | Men | 7.95 | 2.103 | 7.65 | 2.138 | 0.129 | Ma3 = Ef3 |
| | W. | 8.33 | 1.938 | 8.20 | 2.023 | 0.481 | Ma3 = Ef3 |
| 4. Social networks | Men | 5.28 | 3.400 | 4.27 | 2.857 | **0.002 *** | Ma4 > Ef4 |
| | W. | 6.80 | 2.581 | 5.69 | 2.760 | **0.004 *** | Ma4 > Ef4 |
| 5. Resources through web 2.0 applications (blogs, Wikis, Forums, etc.) | Men | 5.23 | 2.919 | 5.25 | 3.062 | 0.955 | Ma5 = Ef5 |
| | W. | 5.59 | 2.765 | 6.46 | 2.494 | **0.007 *** | Ma5 < Ef5 |
| 6. Storage within cloud environments (Google Drive, Dropbox, etc.) | Men | 6.93 | 3.058 | 6.23 | 3.050 | **0.041 *** | Ma6 > Ef6 |
| | W. | 7.67 | 2.379 | 7.43 | 2.520 | 0.425 | Ma6 = Ef6 |
| 7. Social bookmarks and content syndication to share information (Delicious, Mister Wong, FeedReader, etc.) | Men | 1.58 | 2.424 | 2.30 | 2.493 | **0.016 *** | Ma7 < Ef7 |
| | W. | 1.61 | 2.551 | 3.07 | 2.869 | **0.000 *** | Ma7 < Ef7 |
| 8. Management platforms (Moodle, Blackboard, etc.) | Men | 7.88 | 1.606 | 8.03 | 2.277 | 0.536 | Ma8 = Ef8 |
| | W. | 8.46 | 1.467 | 8.39 | 2.304 | 0.809 | Ma8 = Ef8 |
| 9. Device protection software | Men | 4.95 | 3.377 | 3.68 | 3.234 | **0.006 *** | Ma9 > Ef9 |
| | W. | 5.61 | 2.836 | 5.43 | 2.747 | 0.551 | Ma9 > Ef9 |
| 10. Information search databases and thesauri | Men | 7.30 | 1.977 | 7.77 | 2.037 | 0.093 | Ma10 = Ef10 |
| | W. | 7.79 | 1.890 | 7.95 | 2.291 | 0.540 | Ma10 = Ef10 |
| 11. Tools for creating QR codes | Men | 3.33 | 3.467 | 2.82 | 2.765 | 0.296 | Ma11 = Ef11 |
| | W. | 3.56 | 3.139 | 3.84 | 2.740 | 0.477 | Ma11 = Ef11 |
| 12. Personal Learning Environments | Men | 3.70 | 3.201 | 4.87 | 3.372 | **0.000 *** | Ma12 < Ef12 |
| | W. | 4.84 | 2.876 | 5.95 | 2.901 | **0.000 *** | Ma12 < Ef12 |
| 13. Use of ICT in a collaborative way | Men | 5.57 | 3.011 | 6.35 | 3.102 | **0.007 *** | Ma13 < Ef13 |
| | W. | 6.31 | 2.913 | 7.13 | 2.611 | **0.001 *** | Ma13 < Ef13 |
| 14. Preparation of materials through presentations and multimedia | Men | 6.42 | 2.770 | 7.12 | 2.558 | **0.011 *** | Ma14 < Ef14 |
| | W. | 6.90 | 2.827 | 7.59 | 2.571 | **0.009 *** | Ma14 < Ef14 |
| 15. Knowledge of copyright and intellectual property | Men | 4.53 | 3.218 | 6.13 | 2.890 | **0.003 *** | Ma15 < Ef15 |
| | W. | 5.33 | 3.275 | 7.02 | 2.705 | **0.003 *** | Ma15 < Ef15 |

**Table 4.** *Cont.*

|  | | Driving (Ma) | | Effectiveness (Ef) | | Test T | |
|---|---|---|---|---|---|---|---|
| ÍTEMS | Gen. | M | DT | M | DT | *p* | Contrast |
| 16. Bibliographic managers (Zotero, Refworks, etc.) | Men | 4.53 | 3.218 | 5.67 | 3.198 | **0.001*** | Ma16 < Ef16 |
| | W. | 5.33 | 3.275 | 5.95 | 2.177 | 0.137 | Ma16 < Ef16 |
| 17. Effective search and discrimination of relevant information on the web | Men | 6.65 | 3.080 | 7.48 | 2.855 | **0.007 *** | Ma17 < Ef17 |
| | W. | 7.84 | 2.177 | 8.43 | 2.004 | **0.009 *** | Ma17 < Ef17 |
| 18. Online publishing tools (Pinterest, Instagram, SlideShare, etc.) | Men | 3.88 | 3.289 | 4.12 | 3.157 | 0.490 | Ma18 = Ef18 |
| | W. | 5.08 | 3.373 | 5.25 | 3.150 | 0.651 | Ma18 = Ef18 |

Gen. = gender; * $p < 0.05$: significant change.

With regard to attitudes of predisposition to the use of ICT (Table 5), men generally scored highly on all the items. Of particular note is their assessment of the importance of pedagogical updating in ICT for teachers (M = 8.07; SD = 2.007) and their belief that ICTs are tools that enrich the teaching–learning process (M = 7.88; SD = 1.992). In contrast, men believed that ICTs do not encourage pupils' creativity and imagination very much (M = 5.97; SD = 2.435) and that the use of mobile devices does not favour the implementation of emerging technologies (M = 5.90; SD = 2.995). The participating women valued the use of ICTs positively. They highlighted the importance of pedagogical updating in ICTs (M = 8.61; SD = 1.686) and the enrichment that these technologies represented in the teaching–learning process (M = 8.69; SD = 1.323). However, like the participating men, women rated lower the statements which related ICTs with creativity and imagination (M = 7.21; SD = 2.229) and to the use of mobile devices in the classroom as an element that encourages the implementation of emerging technologies (M = 7.18; SD = 2.507).

**Table 5.** Comparisons of attitudes of predisposition in the use of ICTs by gender.

|  | | | | Test T | |
|---|---|---|---|---|---|
| ÍTEMS | Gen. | M | DT | *p* | Contrast |
| 1. The pedagogical update of ICT in the university teaching staff is essential | Men | 8.07 | 2.007 | 0.112 | Mm1 = Mw1 |
| | W. | 8.61 | 1.686 | | |
| 2. ICTs enrich the teaching–learning process | Men | 7.88 | 1.992 | **0.010 *** | Mm2 < Mw2 |
| | W. | 8.69 | 1.323 | | |
| 3. With ICT, learning can occur anytime and anywhere | Men | 7.20 | 2.448 | **0.002 *** | Mm3 < Mw3 |
| | W. | 8.36 | 1.602 | | |
| 4. ICTs promote the creativity and imagination of students | Men | 5.97 | 2.435 | **0.004 *** | Mm4 < Mw4 |
| | W. | 7.21 | 2.229 | | |
| 5. ICTs favour collaborative networking | Men | 7.67 | 2.326 | **0.014 *** | Mm5 < Mw5 |
| | W. | 8.62 | 2.845 | | |
| 6. The use of mobile devices in the classroom encourages the implementation of emerging technologies | Men | 5.90 | 2.995 | **0.012 *** | Mm6 < Mw6 |
| | W. | 7.18 | 2.507 | | |
| 7. The use of ICT increases the motivation of students and the teacher themselves | Men | 6.07 | 2.564 | **0.003 *** | Mm7 < Mw7 |
| | W. | 7.39 | 2.246 | | |
| 8. Although they do not solve all classroom problems, ICTs improve the quality of education | Men | 6.52 | 2.831 | **0.012 *** | Mm8 < Mw8 |
| | W. | 7.69 | 2.172 | | |

Gen. = gender; Mm = mean men; Mw = mean women; * $p < 0.05$: significant change.

Comparing the results for both genders, women rated all statements higher than men and significant differences ($p < 0.05$) were obtained in seven of those statements. The greatest disparity in results occurred in the item that relates ICT with learning anytime and anywhere ($p = 0.002$) and in the item that relates the use of these tools to increased motivation among students and teachers ($p = 0.003$).

Regarding teachers' behaviours in the uses of ICT to support the educational process (Table 6), both men and women obtained very variable scores. In the case of men, only two items exceeded a score of six. The first of these referred to the use of digital content in the classroom (M = 7.73; SD = 2.074) and the second to the structuring of activities using university virtual campuses (M = 6.67; SD = 3.166). Likewise, the technological resources least implemented by male teachers were augmented reality applications (M = 1.67; SD = 2.370), e-portfolios (M = 1.83; SD = 2.585), and social networks (M = 1.97; SD = 3.003). In the case of women, the results showed that they regularly use digital content (M = 8.31; SD = 1.928), activities within virtual campuses, (M = 7.46; SD = 2.964) and videos as classroom materials (M = 6.97; SD = 2.898). Like men, the digital tools least used by women were augmented reality applications (M = 1.92; SD = 2.685), e-portfolios (M = 2.59; SD = 3.170), and social networks (M = 3.25; SD = 3.380).

**Table 6.** Comparison by gender on behaviour in the use of ICT to support the educational process.

| ÍTEMS | Gen. | M | DT | T-test | |
| --- | --- | --- | --- | --- | --- |
| | | | | *p* | Contrast |
| 1. I create learning environments with ICT in the classroom | Male. | 5.73 | 2.609 | **0.012 *** | Mm1 < Mw1 |
| | W. | 6.89 | 2.346 | | |
| 2. I use digital content as support within the classroom | Male. | 7.73 | 2.074 | 0.115 | Mm2 = Mw2 |
| | W. | 8.31 | 1.928 | | |
| 3. I structure subject activities using virtual university campuses | Male. | 6.67 | 3.166 | 0.158 | Mm3 = Mw3 |
| | W. | 7.46 | 2.964 | | |
| 4. I use web tools as subject activities | Male. | 3,77 | 3,407 | 0.145 | Mm4 = Mw4 |
| | W. | 4,67 | 3,375 | | |
| 5. I use applications for the creation of augmented reality as an educational resource in the classroom | Male. | 1.67 | 2.370 | 0.586 | Mm5 = Mw5 |
| | W. | 1.92 | 2.685 | | |
| 6. I use the e-portfolio as an activity for student self-development | Male. | 1.83 | 2.585 | 0.153 | Mm6 = Mw6 |
| | W. | 2.59 | 3.170 | | |
| 7. I use video as classroom material for learning | Male. | 5.90 | 3.530 | 0.072 | Mm7 = Mw7 |
| | W. | 6.97 | 2.898 | | |
| 8. I provide students with ICT tools for planning autonomous learning | Male. | 5.67 | 3.208 | 0.147 | Mm8 = Mw8 |
| | W. | 6.48 | 2.879 | | |
| 9. I evaluate the achievement of subject competencies using ICT tools | Male. | 4.42 | 3.321 | 0.135 | Mm9 = Mw9 |
| | W. | 5.33 | 3.335 | | |
| 10. I use social networks as a resource in the classroom | Male. | 1.97 | 3.003 | **0.030 *** | Mm10 < Mw10 |
| | W. | 3.25 | 3.380 | | |
| 11. I use online questionnaires to assess or detect needs | Male. | 3.85 | 3.584 | **0.001 *** | Mm11 < Mw11 |
| | W. | 6.02 | 3.418 | | |

Gen. = gender; Mm = mean men; Mw = mean women; * $p < 0.05$: significant change.

Comparing the results for both genders, women obtained better scores in all the items. The differences were significant in the cases of the creation of learning environments with ICT ($p = 0.012$), the uses of social networks ($p = 0.030$), and the application of online questionnaires for the evaluation or detection of needs ($p = 0.001$).

Finally, if we look at the results in relation to the effectiveness of ICT in the development of 21st century competencies (Table 7), they were not as variable as in the rest of the sections of the questionnaire. Men believed that technology could be used as a learning tool (M = 6.90; SD = 2.341), that it facilitates students' access to learning materials and content (M = 8.10; SD = 1.920), and that it allows students to work at their own pace (M = 6.73; SD = 2.216). However, men believed that ICT does not favour the development of written and oral expression (M = 4.08; SD = 2.560), nor does it take into account students' strengths, weaknesses, and interests (M = 5.15; SD = 2.550). Women thought that technological tools facilitated access to learning materials and content (M = 8.44; SD = 1.659), allowed students to learn (M = 7.72; SD = 1.916), and promoted autonomy (M = 7.66; SD = 2.032). Nonetheless, they valued less the use of ICT to develop oral and written expression (M = 5.03; SD = 2.732) and critical thinking (M = 6.38; SD = 2.621).

When comparing the results for both genders, the scores for all items were higher in women and that difference was significant ($p < 0.05$) in some cases. The greatest disparity occurred in the belief that ICT is a tool that allows learning from and with peers ($p = 0.000$), that serves for self-assessing ($p = 0.001$), and that takes into account the strengths and interests of the students ($p = 0.002$).

**Table 7.** Comparison of competencies for the 21st century by gender.

| ÍTEMS | Gen. | M | DT | Test T | |
| --- | --- | --- | --- | --- | --- |
| | | | | *p* | Contrast |
| 1. Make materials more flexible | Men | 6.40 | 2.592 | 0.118 | Mm1 = Mw1 |
| | W. | 7.07 | 2.024 | | |
| 2. Allow students to work at their own pace | Men | 6.73 | 2.216 | **0.022 *** | Mm2 < Mw2 |
| | W. | 7.57 | 1.756 | | |
| 3. Facilitate fun and learning | Men | 5.85 | 2.815 | **0.003 *** | Mm3 < Mw3 |
| | W | 7.18 | 1.979 | | |
| 4. Increase student motivation | Men | 6.00 | 2.792 | **0.021 *** | Mm4 < Mw4 |
| | W. | 7.07 | 2.205 | | |
| 5. Allow students to collaborate with their classmates | Men | 6.65 | 2.530 | 0.112 | Mm5 = Mw5 |
| | W. | 7.31 | 1.979 | | |
| 6. Allow students to learn from/with their peers | Men | 5.83 | 2.485 | **0.000 *** | Mm6 < Mw6 |
| | W. | 7.41 | 1.944 | | |
| 7. Make it easy for students to access learning materials and content | Men | 8.10 | 1.920 | 0.295 | Mm7 = Mw7 |
| | W. | 8.44 | 1.659 | | |
| 8. Develop oral and written expression | Men | 4.08 | 2.560 | 0.051 | Mm8 = Mw8 |
| | W. | 5.03 | 2.732 | | |
| 9. Enable students to learn with the use of technology | Men | 6.90 | 2.341 | **0.037 *** | Mm9 < Mw9 |
| | W. | 7.72 | 1.916 | | |
| 10. Take into account the strengths, weaknesses, and interests of students | Men | 5.15 | 2.550 | **0.002 *** | Mm10 < Mw10 |
| | W. | 6.44 | 1.831 | | |
| 11. Propose a facilitating learning climate | Men | 5.98 | 2.759 | **0.003 *** | Mm11 < Mw11 |
| | W. | 7.28 | 1.845 | | |
| 12. Allow students to participate in decision-making | Men | 5.63 | 2.477 | 0.078 | Mm12 = Mw12 |
| | W. | 6.44 | 2.527 | | |

**Table 7.** *Cont.*

| | | | | Test T | |
| ÍTEMS | Gen. | M | DT | p | Contrast |
|---|---|---|---|---|---|
| 13. Allow students to participate in problem solving | Men | 5.97 | 2.379 | 0.073 | Mm13 = Mw13 |
| | W. | 6.74 | 2.309 | | |
| 14. Develop your critical thinking | Men | 5.42 | 2.533 | **0.043 \*** | Mm14 < Mw14 |
| | W. | 6.38 | 2.621 | | |
| 15. Self-assess your learning progress | Men | 6.23 | 2.382 | **0.001 \*** | Mm15 < Mw15 |
| | W. | 7.56 | 1.996 | | |
| 16. Promote autonomy in their learning | Men | 6.63 | 2.131 | **0.008 \*** | Mm16 < Mw16 |
| | W. | 7.66 | 2.032 | | |
| 17. Improve your learning processes | Men | 6.32 | 2.303 | 0.056 | Mm17 = Mw17 |
| | W. | 7.10 | 2.150 | | |
| 18. Increase student creativity | Men | 5.65 | 2.413 | 0.070 | Mm18 = Mw18 |
| | W. | 6.48 | 2.547 | | |

Gen. = gender; Mm = mean men, Mw = mean women; * $p < 0.05$: significant change.

## 4. Discussion

This study shows a differentiated profile of ICT-use among university professors of Social and Legal Sciences depending on whether they are men or women. This statement is supported by the data provided in the present study, in which women have a higher score in the use of 16 of the 18 ICT tools asked about in the questionnaire. This difference between men and women is statistically significant in four of them. These results allow us to conclude that the gender variable conditions of the use of ICT tools in university classrooms are contrary to the conclusions of studies carried out some years ago, such as that of Onasanya et al., [53] in which the authors indicated that the gender of the teachers did not condition the integration of ICT in the classroom.

In the same way, this study suggests a greater mastery of ICT by women university professors, except for the use of word processors, spreadsheets, and basic ICT components (storage, peripheral elements, etc.), in which men are more skilled. At this point, it should be noted that these statements are based on the collection of the teachers' perceptions and may not be an accurate reflection of what happens in reality. After an exhaustive review of research related to ICT management, it can be pointed out that some studies suggested a greater mastery for men in the management of ICT tools [44–46], and therefore, they are far from the results obtained in the present study. However, some researchers such as Vázquez-Cano et al. [43] indicated that, currently, both men and women are equally competent in the different ICT tools. Likewise, studies carried out in the last decade, such as those of García-Valcárcel and Arras [46] and Araiza and Pedraza [47], shared with the present study the break with traditional ideas such as that of men having a greater command of computer tools than women. In all two studies, the evidence suggested greater knowledge and use of ICT tools by women.

Regarding the degree of effectiveness of ICT in teaching–learning processes at university level, if we compare the results of men and women professors in the areas of Social Sciences and Law, this study reveals that the degree of effectiveness in the use of ICT in the teaching–learning process is higher among women than among men. In six of the ICT tools, the difference is statistically significant. As in Araiza and Pedraza's research [47], the women participating in the present study expressed a better assessment of their teaching skills with ICTs and their incorporation in the teaching–learning process than men. In this way, they integrate these tools in the disciplinary subjects to promote educational

innovation, improve the assessment of student performance, and make their lessons more dynamic in order to increase student participation.

As for the use of ICT as a support in the educational process by the university teachers in the present study, significant differences were found between men and women, with women obtaining the highest scores. These higher scores in women teachers are supported by Area [54], Mirete [27], and Araiza and Pedraza [47]. Furthermore, the present study shares with Araiza and Pedraza's research the suggestion that women teachers are more likely than men teachers to believe that the use of ICT increases the motivation of both teachers and students.

## 5. Conclusions

In conclusion, and in response to the specific objectives of this study (specifically the first one) it should be noted that the use of ICT is variable in both genders among university teachers in the area of Social and Legal Sciences, with different levels of mastery of the various tools. Women perceive greater self-efficacy in the use of technologies than men. In other words, women generally have a higher perception of self-management than men in all ICT tools and make greater use of social networks.

In relation to the second specific objective of the study (related to the teachers' valuation of technologies as educational support), we can affirm that gender is a determining factor in the use of ICT in education, as reflected in the perception that the participants in the study have on the use of ICT as a support for the educational process. Both men and women understand that ICTs enrich the teaching–learning process, but women perceive a greater potential in the uses of ICTs and their applications in teaching, while men have a more traditional view of which tools are more effective for learning. In this case, men make less use of digital content and activities, while women incorporate more of these types of content and activities in the classroom. In both cases, there is still work to be done in relation to the introduction and use of enhanced reality, the e-portfolio, and social networks in the classroom. For all these reasons, we can state that gender determines the use of ICT tools that are considered more effective in teaching–learning processes.

With regard to the specific objective of analysing the effectiveness of ICT in the development of 21st century skills and seeing if there are differences by gender, we can say that both genders value it positively, but with nuanced differences. Teachers have different opinions on the effectiveness of using ICT for the development of 21st century skills (collaboration, critical thinking, and creativity, etc.). Men perceive that they help by facilitating access to materials and content and that they allow students to work at their own pace. However, they do not support the use of ICT for the development of written and oral expression, nor do they think that ICTs take into account students' strengths, weaknesses, and interests. Women agree with regard to these weaknesses and also include that ICTs do not favour the development of critical thinking. They do understand that ICTs facilitate learning and autonomy. The greatest disparity between genders is found in the belief that ICT (as a tool) allows learning with others and facilitates collaborative learning—which women value more positively.

University teachers, both men and women, value positively the need for and the importance of pedagogical updating in ICTs. Because, in order for university students to develop these generic competences, universities should support the implementation of teaching and learning strategies that train in their use and that also serve for the development and acquisition of other competences [55,56]. To this end, it is necessary to provide more training in digital competence for teachers [57,58]. All this is key to responding to the current pandemic society and to the great technological challenges that must be faced on a daily basis as a result of it. These circumstances are transforming our society by accelerating the generation of increasingly digital contexts that require new strategies and skills.

In order to initiate the necessary training in digital competencies, the educational environment can be a suitable space to develop good teaching practices with ICTs and

drive education towards SDG 2030. The inclusion of sustainability principles in university classrooms will improve the quality of education while experimenting in a sustainable use of technology [59].

Finally, with respect to the limitations of the study, this research could have had a larger sample. It could also have looked more deeply into aspects such as the relationship between the teachers' years of experience and their perception of self-efficacy in ICT-use, or their assessment of ICT as a tool to support education. Moving on to possible future lines of research, it might be interesting to carry out this same analysis with a larger sample of teachers from different branches of knowledge. By doing this, a more global and complete vision could be obtained, assessing the possible differences between all these different professionals.

**Author Contributions:** Conceptualization, S.C.B., M.L.O. and S.V.T.; methodology, S.C.B. and C.L.C.; validation, S.A.O. and M.L.O.; formal analysis, S.C.B. and S.V.T.; resources, M.L.O.; data curation, C.L.C. and S.A.O.; writing—original draft preparation, S.C.B., M.L.O., S.V.T., C.L.C. and S.A.O.; writing—review and editing, S.C.B., M.L.O., S.V.T., C.L.C. and S.A.O.; supervision, S.C.B. and M.L.O. All authors have read and agreed to the published version of the manuscript.

**Funding:** This research received no external funding.

**Institutional Review Board Statement:** This research complies with our university's Code of Good Research Practice in accordance with the National Statement of Scientific Integrity. (http://www.unizar.es/actualidad/ficheros/20181114/44451/codigo_de_buenas_practicas_en_investigacion_aprobado_cdg.pdf (accessed on 10 January 2021)). The ethical review and approval of this study by the Ethics Committee of our autonomous community, the Aragonese Institute of Health Sciences, IACS, has been waived, given that it belongs to the area of Biomedicine and Health Sciences (health), a field far removed from the social and legal sciences to which this study belongs.

**Informed Consent Statement:** Informed consent was obtained from all subjects involved in the study.

**Data Availability Statement:** Not applicable.

**Conflicts of Interest:** The authors declare no conflict of interest.

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
