# Peer review of "Digital Competence of University Teachers of Social and Legal Sciences from a Gender Perspective"

_education, doi:10.3390/educsci11120806_

Round 1
Reviewer 1 Report
After analyzing the manuscript carefully we can say that it is a very interesting and well developed research. It is very well structured and well thought out. The bibliography used is very updated and well referenced throughout the manuscript.
The central theme of the research is very interesting, since it not only focuses on the importance of technologies and ICT in education, but also takes into account the possible difference by gender. This is something very relevant today and on which few studies focus.
In order to comment on any possible improvement, it can be observed how the authors set out 3 general objectives, which are very ambitious and well structured; however, it could be beneficial for the study and for the reader's understanding to create specific objectives for each of the general objectives.
The results of the study are well described. They are well differentiated by sections and very well supported by different tables, very visual. The discussion is very well presented and referenced by comparing it with similar very recent studies, as is the case of one used from 2010, comparing it with the one in the manuscript (2021). It can be seen in the comparison how the results obtained are very different and the studies performed are very similar. It is also worth noting the reference to the SDG 2030 in the conclusions section as a very ambitious aspect of the manuscript.
Author Response
Please see the attachment
COVER LETTER
We would like to thank the professionals who carried out the review for all the contributions and recommendations they have made. They will undoubtedly result in an improvement of the work presented.
Before detailing the modifications introduced, we would like to point out that all the observations made by both reviewers have been carefully considered with a high level of self-demand.
The modifications based on the reviewers' suggestions are as follows:
Reviewer 1:
|
Nº |
Reviewer’s comments |
Authors’ replies
|
|
1 |
it could be beneficial for the study and for the reader's understanding to create specific objectives for each of the general objectives. |
Specific objectives have been added on page 3. Lines 144-150
|

Reviewer 2 Report
- The significance of the research SHOULD be elaborated in detail in the introduction part; the gap between the current research and prior findings should also be clarified.
- Please add hypotheses of the study
- Elaborate on the population of the study.
- The normality procedures of the data need to be informed
- The recommendation and limitations of the study are still limited; thus they need to be more informed in detail.
Author Response
Please see the attachment
COVER LETTER
We would like to thank the professionals who carried out the review for all the contributions and recommendations they have made. They will undoubtedly result in an improvement of the work presented.
Before detailing the modifications introduced, we would like to point out that all the observations made by both reviewers have been carefully considered with a high level of self-demand.
The modifications based on the reviewers' suggestions are as follows:
To Reviewer:
|
Nº |
Reviewer’s comments |
Authors’ replies
|
|
1 |
The significance of the research SHOULD be elaborated in detail in the introduction part; the gap between the current research and prior findings should also be clarified.
|
This aspect has been developed in lines 104-107 and 124-126. |
|
2 |
Please add hypotheses of the study |
The hypotheses of the study have been added on page 3. Lines 128-133 |
|
3 |
Elaborate on the population of the study |
Information about the population has been added on page 4. |
|
4 |
The normality procedures of the data need to be informed |
The normality procedures have been described. The procedure for contacting research participants has been specified in the procedure section. All this information has been added on page 5.
|
|
5 |
The recommendation and limitations of the study are still limited; thus, they need to be more informed in detail |
The recommendations and limitations of the study have been added on page 13. |

Reviewer 3 Report
This paper aims to show the differences between men and women professors’ attitudes, beliefs and self-perceptions on their management and effectiveness of ICT use in teaching. Its main contributions are the introduction of a gender perspective and findings that there are differences not previously identified, and the inclusion of social and legal sciences as a discipline. The article builds on existing work in an interesting way to set up the discussion.
The manuscript is generally well-structured, though the arguments lack some clarity and are sometimes difficult to follow. A general comment is that the meaning of the words competence, management and effectiveness are not sufficiently explained in the context of the study. Is there a specific framework (ie, a list of actual tasks an individual can perform) or is it purely a generic use of the word and therefore open to interpretation? I think that this has consequences for the development of the discussion around differences in skills between men and women.
There are also some more detailed comments that I have given some examples of this in the specific comments below.
My background does not qualify me to comment in any detailed way on the design of the study, statistical analysis methods, etc. Overall, the description and broad reasoning are clear, and the method seems appropriate for addressing the research objectives. It would be helpful to have some further explanation as regards the way in which the sample participants were reached, eg, was recruitment via an existing mail list or particular network? In addition, there is no mention of ethical approval. The tables are easy to follow, but on some occasions have been incorrectly reported upon in the text. I have picked out some of these and shown in the specific comments, though all the figures used in the text should be re-checked.
The references are generally appropriate, though several the references used for the discussion, eg, [43]-[45], [52], [54] are at least ten years old. If it is not possible to draw on more recent work in this area, there should be explicit acknowledgement of this.
The conclusion sums up the findings satisfactorily, however, the final pulling together of the arguments does not sufficiently explain the value of understanding gender differences in meeting the desired goals. The introduction of SDG 2030 at this late stage also shows a lack of cohesion in the overall argument.
Specific comments
Line 47 – I found this paragraph confusing, as it does not explain clearly what is meant by ‘digital competence’ and its role. This is a good opportunity to clarify at an early stage in the article what is meant by competence.
Line 63 - The sentence beginning ‘In this sense…’ is unclear and again presents a good opportunity to clarify the definition of digital competence to feed into later discussion.
Line 72 – does this refer to frequency of use? This isn’t clear.
Line 77 – the phrase ‘on the other hand’ is a colloquial one and is used several times (along with ‘on one hand’) throughout the article. I suggest that these are replaced.
Line 77 – I was not sure what this paragraph contributes to the overall argument: does the fact that students actively participating in their own learning have any impact on the overall conclusions?
Line 102 – check here and throughout for use of the words gender, sex, male, female, women and men. ‘Female’ and ‘male’ are categories of sex and are not a gender. The study should either be presented as categorised according to sex (ie male/female) or gender (men/women).
Line 118 – its unclear what ‘management’ and ‘effectiveness’ mean in this context – a definition of this earlier in the paper would increase its clarity and it’s important that the reader understands how these are being used for the purposes of later discussion.
Line 179 – the figure M=1.61 is incorrectly reported (it’s the figure reported as female for this category under table 2). Double-check all the reported figures highlighted in the results.
Line 220 – I became a little unclear at this point which of the three ‘objectives’ (outlined from line 114) is being addressed. Recommend that the results section is ordered according to objectives 1, 2 and 3 to make it easy to follow, and that each section is clearly signposted according to the objective that it meets.
Line 263 – the figure M=6.97 is incorrectly reported (it’s the figure reported as female for this category under table 6). Double-check all the reported figures highlighted in the results.
Line 307 – there is some discussion of ‘mastery’ and ‘more skilled’ in this section which may be overstating what can be known about the individuals according to their gender. The study is based on the perceptions of the participants, ie, the study is not an audit of their actual skills. The perceptions of individuals may differ from reality (and this may be influenced by their gender), and this should at least be acknowledged somewhere in the discussion.
Line 357 – it’s not clear how this conclusion ties in with the gendered findings that you discuss in the previous section. How do the findings inform this conclusion? What are the limitations of the findings?
Author Response
Please see the attachment
COVER LETTER
We would like to thank the professionals who carried out the review for all the contributions and recommendations they have made. They will undoubtedly result in an improvement of the work presented.
Before detailing the modifications introduced, we would like to point out that all the observations made by both reviewers have been carefully considered with a high level of self-demand.
The modifications based on the reviewers' suggestions are as follows:
To Reviewer:
|
Nº |
Reviewer’s comments |
Authors’ replies
|
|
1 |
A general comment is that the meaning of the word’s competence, management and effectiveness are not sufficiently explained in the context of the study. Is there a specific framework (ie, a list of actual tasks an individual can perform) or is it purely a generic use of the word and therefore open to interpretation? I think that this has consequences for the development of the discussion around differences in skills between men and women.
|
It is a purely generic use of the word.
|
|
2 |
It would be helpful to have some further explanation as regards the way in which the sample participants were reached, eg, was recruitment via an existing mail list or particular network? |
The explanation has been added on lines 159-164 (Participants and Sample section). |
|
3 |
there is no mention of ethical approval. |
"This research complies with our university's Code of Good Research Practice in accordance with the National Statement of Scientific Integrity.
You will find the link to the code of good practice at the end of this cover letter. |
|
4 |
The tables are easy to follow, but on some occasions have been incorrectly reported upon in the text. I have picked out some of these and shown in the specific comments, though all the figures used in the text should be re-checked.
|
The results and tables have been revised. |
|
5 |
The references are generally appropriate, though several the references used for the discussion, eg, [43]-[45], [52], [54] are at least ten years old. If it is not possible to draw on more recent work in this area, there should be explicit acknowledgement of this.
|
The authors of these sources are leading figures in the field, and there is little research on the use of ICTs in education in relation to gender. |
|
6 |
the final pulling together of the arguments does not sufficiently explain the value of understanding gender differences in meeting the desired goals. The introduction of SDG 2030 at this late stage also shows a lack of cohesion in the overall argument. |
This has already been referenced on page 2 of the article. Lines 47-50 |
|
7 |
Line 47 – I found this paragraph confusing, as it does not explain clearly what is meant by ‘digital competence’ and its role. This is a good opportunity to clarify at an early stage in the article what is meant by competence |
Further information has been added on page 2, as well as a reference (lines 47-50). |
|
8 |
Line 63 - The sentence beginning ‘In this sense…’ is unclear and again presents a good opportunity to clarify the definition of digital competence to feed into later discussion.
|
The definition has been reworded at the beginning of the previous paragraph, at the end of the section. The wording has been improved. (Lines 69-73) |
|
9 |
Line 72 – does this refer to frequency of use? This isn’t clear.
|
Yes, it does refer to the frequency of use. |
|
10 |
Line 77 – the phrase ‘on the other hand’ is a colloquial one and is used several times (along with ‘on one hand’) throughout the article. I suggest that these are replaced.
|
Modifications related to this have been made throughout the text. |
|
11 |
Line 77 – I was not sure what this paragraph contributes to the overall argument: does the fact that students actively participating in their own learning have any impact on the overall conclusions?
|
Active learning and involvement in active learning can be promoted by the implementation and use in the teaching-learning methodology of ICT, so this argument is introduced in the content. |
|
12 |
Line 102 – check here and throughout for use of the words gender, sex, male, female, women and men. ‘Female’ and ‘male’ are categories of sex and are not a gender. The study should either be presented as categorised according to sex (ie male/female) or gender (men/women).
|
We have modified this aspect in the text. We now only use the terms associated with gender (women and men). |
|
13 |
Line 118 – it’s unclear what ‘management’ and ‘effectiveness’ mean in this context – a definition of this earlier in the paper would increase its clarity and it’s important that the reader understands how these are being used for the purposes of later discussion.
|
A definition of both concepts has been specified in the "data collection instrument" section. Lines 180-183 |
|
14 |
Line 179 – the figure M=1.61 is incorrectly reported (it’s the figure reported as female for this category under table 2). Double-check all the reported figures highlighted in the results.
|
The results and tables have been revised |
|
15 |
Line 220 – I became a little unclear at this point which of the three ‘objectives’ (outlined from line 114) is being addressed. Recommend that the results section is ordered according to objectives 1, 2 and 3 to make it easy to follow, and that each section is clearly signposted according to the objective that it meets. |
The results derived from this study have been structured around the six dimensions of analysis defined in the methodological framework. We understand that much of the success in analysing, structuring and clarifying the research depends on the establishment of clear dimensions or categories throughout the process.
|
|
16 |
Line 263 – the figure M=6.97 is incorrectly reported (it’s the figure reported as female for this category under table 6). Double-check all the reported figures highlighted in the results.
|
The results and tables have been revised |
|
17 |
Line 307 – there is some discussion of ‘mastery’ and ‘more skilled’ in this section which may be overstating what can be known about the individuals according to their gender. The study is based on the perceptions of the participants, ie, the study is not an audit of their actual skills. The perceptions of individuals may differ from reality (and this may be influenced by their gender), and this should at least be acknowledged somewhere in the discussion.
|
An explanatory sentence has been added specifying that these results are based on the personal perceptions of the participants. Lines 347-349 |
|
18 |
Line 357 – it’s not clear how this conclusion ties in with the gendered findings that you discuss in the previous section. How do the findings inform this conclusion? What are the limitations of the findings? |
The conclusions have been structured according to the objectives, relating these conclusions to the gendered results.
Limitations and new lines of research emerging from the study have been added. (Lines 423-430) |

Round 2
Reviewer 3 Report
I am happy that the comments raised have all been addressed satisfactorily.